# Dynamically weighted graph neural network for detection of early mild cognitive impairment

**Li Liu[1,2], Yifei Li[3], Kai Yang[ID][4]***

**1** School of Science, China Pharmacy University, Nanjing, China, **2** Hainan University, Haikou, China, **3** Department of Mechanical Engineering, Huzhou University, Huzhou, China, **4** School of Mathematics and Information Science, Nanjing Normal University of Special Education, Nanjing, China

* yangkai@njts.edu.cn

**Data availability statement:** The data underlying the results presented in the study are available from ADNI database (https://adni.loni.usc.edu).

## Abstract

Alzheimer's disease (AD) is a prevalent neurodegenerative disease that primarily affects the elderly population. The early detection of mild cognitive impairment (MCI) holds significant clinical importance for prompt intervention and treatment of AD. Currently, functional connectivity (FC) networks-based diagnostic methods for early MCI (eMCI) detection are widely employed. FC considers the interaction patterns between brain regions as a topological structure. Recently, graph neural network (GNN) approaches have been utilized for disease diagnosis using FC networks, leveraging their ability to extract features from topological structures. However, existing methods typically treat FC features as GNN node attributes, disregarding the fact that FC features represent the weights of connection edges in FC networks with topological characteristics. In this paper, we propose a dynamically weighted GNN-based approach for early eMCI detection. Our method takes into account the topological structure and dynamic properties of FC networks by utilizing temporal FC features as the weighted adjacency matrix for dynamic GNNs. Moreover, the weighted graph local clustering coefficient of brain regions is employed as the node feature for GNNs. We extensively evaluated our approach using the ADNI database and achieved accuracies of 91.67% and 78.33% on low- and high-order FC networks, respectively. These results demonstrate the effectiveness and superiority of our proposed method compared to existing approaches.

## Introduction

Alzheimer's disease (AD) is an irreversible neurological disease, which often occurs in aged group. The fundamental clinical symptoms of AD include some momentous cognitive and intellectual impaired brain abilities, such as memory and reasoning skills. This disease will gradually worsen over time, seriously reduces the patients' life quality and eventually leads to death. To date, there is no effective clinical treatment. Therefore, early detection and timely intervention are of great significance to improve the life quality of patients [1,2]. Mild cognitive impairment (MCI) is a condition characterized by mild cognitive decline, which is the middle stage of brain cognitive decline between AD and normal aging. People with MCI have

**Funding:** The author(s) received no specific funding for this work.

**Competing interests:** The authors have declared that no competing interests exist.

a high probability of developing into AD, and the average conversion rate is as high as 10% - 15% per year. The detection of early MCI (eMCI) has important clinical significance for reducing the risk of conversion to AD by providing proper treatment and intervention. Therefore, the accurate detection of eMCI has attracted extensive attention of researchers [1,2].

At present, scholars have developed many methods to diagnose eMCI [3–5]. Among many methods, the functional connectivity (FC) networks based on resting state functional magnetic resonance imaging (RS-fMRI) data are widely used to extract classification features. RS-fMRI, as a noninvasive neuroimaging technology, uses blood oxygen level-dependent (BOLD) signals to measure spontaneous brain functional activities. Because BOLD signal is sensitive to spontaneous and inherent neural activity in the brain, and RS-fMRI has high spatial resolution, RS-fMRI data has been successfully used to construct FC networks for AD/eMCI detection [6,7]. The FC refers to the temporal correlation of BOLD signals between/among different brain regions, which can show how brain regions with structural separation and functional specialization interact [8]. In literatures [9–11], FC networks are typically modeled using topology, that is, the brain region is regarded as a node of the network, and FC is the connection weight between two nodes (brain regions). Due to pathological attacks, there are differences in topology and connection strength between normal FC network and interrupted FC network, so detection of abnormal FC is of great significance to diagnosis of AD/eMCI. In general, researchers extract features from FC networks and feed them into classifiers such as support vector machine and deep neural network for classification.

In the past decades, deep neural networks have been widely used in numerous fields because of its powerful abstract feature extraction ability. Among them, deep neural networks are also employed for FC feature classification, especially the graph neural networks (GNNs) are used to diagnose neurological diseases, such as AD and autism [12,13]. GNN is a method of learning graph data developed in recent years, which is suitable for graphs with topology and is applied to recommendation system, drug discovery [14], disease prediction and other fields [15]. Because both FC networks and GNNs have the characteristics of graph, it is a new possibility to combine FC networks and GNNs for disease diagnosis. However, the existing GNNs usually take FC features as the node features of GNNs. For instance, Zhao et al. [12] proposed a FC feature enhancement method based on graph convolution network, which combines low- and high-order FC features to classify autism. Chen et al. [13] proposed a graph convolution network model with confidence learning, which classifies nodes in graph data with label noise using FC features as node features of the GNN. As mentioned earlier, FC is the weight of the connecting edge of the brain area in the brain network structure. Therefore, although the FC feature as the node feature of GNNs has achieved excellent classification results, it separates the attributes of FC network and GNNs in topology, that is, the edge attribute in FC network is regarded as the node feature in GNNs.

In this paper, we propose a dynamic GNN method based temporal FC for eMCI diagnosis, which utilizes the edge characteristics of temporal FC in dynamic brain network. Specifically, we propose to use temporal FC feature as the weighted edge of the dynamic GNN, and use the weighted graph local clustering coefficient [16] as the node feature of the dynamic GNN to classify eMCI vs. normal control (NC). For this reason, we investigated two types of dynamic FC networks, namely low-order dynamic FC networks [10,17] and high-order dynamic FC networks [18]. The main difference between different FC networks is the interaction pattern defined between/among the brain regions and the network structure. After establishing two dynamic brain networks, considering that there are redundant connections or connections that are not important for disease classification in the FC network [1,10], we use two-sided $t$-test to select FC features and eliminate redundant features. Then, the Long

Short-Term Memory (LSTM) architectures are adopted to capture latent temporal features for eMCI classification.

- *First*, the FC value in dynamic FC network is regarded as the weighted adjacency matrix in dynamic GNN, which realizes the unification of FC features and GNN in topology.
- *Second*, the proposed model is simple, effective and easy to implement.
- *Third*, the proposed method can be easily extended to other neurological disorders, such as autism spectrum disorder.
- *Finally*, systematic experiments were carried out on large-scale ADNI data sets. The results prove that the suggested method has high discrimination ability for eMCIs and NCs.

## Related works

### Functional connectivity networks

Some neurological diseases are often accompanied by functional abnormalities, such as AD [19–21] and autism [22–24]. In the past decade, researchers and institutions have made great efforts to explore functional biomarkers of neurological diseases [25–27]. Wee et al. [25] proposed a dynamic FC network that utilizes interrupted temporal network particularities to improve the performance of control-patient classification. Liu et al. [26] used multivariable pattern analysis to study the contribution of connectivity subsets between the whole brain FC's to the severity of autism. Zhang et al. [27] proposed a high-order FC network that interacts across frequencies based on low-order FC subnetworks constructed in different frequency bands. Xie et al. [17] proposed a high order FC network, which is based on the center distance characteristics of dynamic FC networks. These methods have made important contributions to building FC features, and many efforts have been made to discover disease biometrics based on classical machine learning methods (e.g., SVM).

Recently, due to the excellent ability of deep neural networks (DNNs) to extract potential features, researchers have shifted their attention to using DNNs to extract features from FC networks for diagnosis of neurological diseases [2–4,9]. Zhang et al. [2] proposed an eMCI diagnosis method based on a few-shot learning strategy that utilizes the similarity between paired FCs among subjects. Wang et al. [3] proposed a spatiotemporal convolutional recurrent neural network for automatic prediction of AD progression and detection of AD related brain regions from RS-fMRI time series. Kam et al. [4] proposed a deep learning framework based on multi paired static and dynamic FC networks to learn the deep embedded spatial patterns of static and dynamic FC networks for eMCI diagnosis.

### Graph neural networks

The graph is a powerful way to denote entities with arbitrary relational structures [28] that coincide with brain FC networks. Several diagnosis methods based on GNN and FC network are proposed [12,29–31]. Zhao et al. [12] proposed a multi-view brain FC network feature enhancement method based on GNN, which enhances node features through the connection relationship between different nodes, and extracts deeper and more discriminative features. Kim et al. [29] proposed a GNN based model for predicting the prognosis of AD using longitudinal neuroimaging data, which uses a temporal FC network as input. Wang et al. [30] defined a functional connectivity based graph and a connectivity based graph convolutional network architecture for fMRI analysis, which can extract spatial features from connected group neighborhoods, consistent with the functional organization of the brain.

Salim and Hamza introduced a graph convolutional aggregation (GCA) model, which applies skip connections and identity mapping to enhance graph node feature learning [32].

This method integrates imaging and non-imaging features into node and edge representations, respectively, achieving improved classification accuracy for autism spectrum disorder (ASD) and Alzheimer's disease. However, its main limitation lies in its reliance on static functional connectivity graphs, which may not fully capture the temporal evolution of brain networks in neurodegenerative diseases. Similarly, Jiang et al. presented a hierarchical graph convolutional network (hi-GCN) that considers both individual brain network topology and group-level associations to improve classification performance [33]. The model effectively learns hierarchical representations of brain networks, but it does not model the temporal dynamics of functional connectivity, which are crucial for detecting eMCI.

In contrast, our proposed Dynamically Weighted GNN integrates temporal FC networks as dynamically weighted adjacency matrices, ensuring that the model captures the evolving interactions between brain regions. Additionally, we employ an LSTM-based sequential model to further refine temporal feature extraction, enabling more accurate eMCI classification.

Hence one can see that GNN has a promising future in the diagnosis of neurological diseases.

## Materials and methods

In this section, we will introduce the method proposed in this article, including constructing a dynamic functional connection network and constructing a dynamic graph neural network.

### Overview of the proposed framework

The framework of the proposed method is shown in Fig 1. It visualizes the methodological pipeline employed within the research to detect eMCI by harnessing the dynamic nature of functional connectivity patterns as discerned from resting-state fMRI data:

(a) Sliding Windows: This segment of the framework delineates the application of a sliding window technique to the resting-state fMRI time series. Each trajectory represents the BOLD signal from a distinct cerebral region. The window length ('w') and step size ('s') are parameters that govern the segmentation of the time series into overlapping intervals, facilitating the capture of dynamic brain connectivity.

(b) Construction of Dynamic FCN: Following the temporal segmentation, this phase entails the construction of a series of functional connectivity networks (FCNs). Each FCN is a snapshot of the brain's functional connectivity at a given time slice, with edges denoting the connectivity strength between pairs of regions, captured across successive time points.

(c) Constructing Dynamic GNN: Transitioning from FCNs to a graph-based representation, this step maps the FCNs onto a graph structure where nodes symbolize the brain regions and weighted edges encapsulate the connectivity strength, denoted by $FC_{ij}(k)$, at each window 'k'. This graph-theoretic approach permits the leveraging of topological information inherent in the brain's functional layout.

(d) RNN Architecture: The sequence of graphs derived from consecutive time windows is processed via a recurrent neural network (RNN). Here, the RNN is tasked with detecting and encoding temporal dependencies across the graph sequence. The $h_k$ states correspond to the hidden layers of the RNN, ideally capturing the evolution of connectivity patterns over time.

(e) Classifier: The final output of the RNN, which aggregates the temporal evolution of the functional connectivity graph features, serves as input to a classifier. The classifier's objective is to discern patterns indicative of eMCI, as opposed to those characteristic of a cognitively normal aging process.

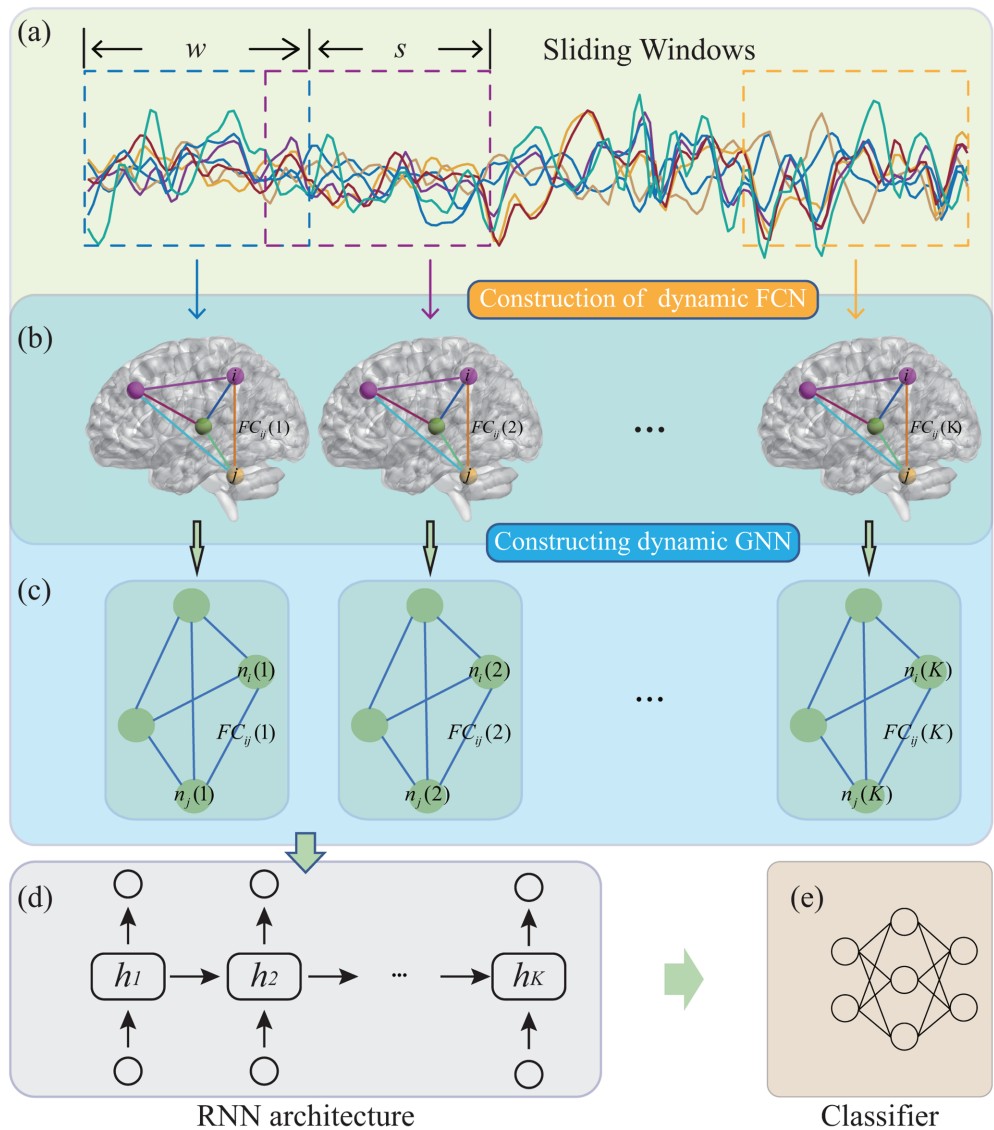

**Fig 1. Overview of the proposed framework.**

In essence, this paper articulates a sophisticated approach that integrates both spatial and temporal dimensions of brain activity, employing advanced neural network architectures to address the complex challenge of early eMCI detection from neuroimaging data.

## Constructing dynamic functional connectivity networks

Functional connectivity reveals the interaction pattern between brain regions by using the temporal correlation of BOLD signals in different brain regions, which is of great significance for discovering functional tissues of human brain and finding biomarkers of neuropsychiatric diseases. Here, we introduce two types of FC networks: dynamic low-order FC network and dynamic high-order FC network.

A subject's RS-fMRI data is denoted as $\mathbf{X} = (x_1, \cdots, x_i, \cdots, x_R) \in \mathbb{R}^{R \times M}$, where $\mathbf{x}_i \in \mathbb{R}^M$. To capture the dynamic interactions between brain regions, we use the sliding window strategy (see Fig 1(a)) to create low-order dynamic FC networks. This involves dividing the RS-fMRI time series into $K = \lfloor M - L \rfloor / s + 1$ overlapping segments using a fixed-length sliding window. Here, $L$ represents the window length and $s$ is the translational step size. Each segment corresponds to a sub-series, denoted as $\mathbf{x}_i(k)$ from the $i$-th brain region within the $k$-th window. Subsequently, we compute the short-time correlation between the $i$-th and $j$-th brain regions for each segment.

$$
\begin{aligned}
lFC_{ij}(k) &= \frac{\langle \mathbf{x}_i(k), \mathbf{x}_j(k) \rangle}{|\mathbf{x}_i(k)||\mathbf{x}_j(k)|} \\
&= \frac{\sum_{t=1}^{M} (x_{it}(k) - \bar{\mathbf{x}}_i(k)) (x_{jt}(k) - \bar{\mathbf{x}}_j(k))}{\sqrt{\sum_{t=1}^{M} (x_{it}(k) - \bar{\mathbf{x}}_i(k))^2} \sqrt{\sum_{t=1}^{M} (x_{jt}(k) - \bar{\mathbf{x}}_j(k))^2}}
\end{aligned}
\tag{1}
$$

where $M = 137$ is the longest length of the RS-fMRI scanning series. Thus, we can get the temporal low-order FC network $\mathbf{lFC}(k) = \left\{ lFC_{ij}(k) \right\}_{1 \leqslant i,j \leqslant R}$ at $k$-th window. Than, the low-order dynamic FC network is denoted as $\mathbf{lFC} = [lFC(1), lFC(2), \cdots, lFC(K)]$, as shown in Fig 1(b).

Let $\mathbf{h}_i(k) = (lFC_{i1}(k), lFC_{i2}(k), \cdots, lFC_{iR}(k))$ represent the $i$-th column of the adjacency matrix of the low-order dynamic FC network. Then, the high-order FC is calculated as follow:

$$
hFC_{ij}(k) = \frac{\langle \mathbf{h}_i(k), \mathbf{h}_j(k) \rangle}{|\mathbf{h}_i(k)||\mathbf{h}_j(k)|}
\tag{2}
$$

Thus, the temporal high-order FC network at $k$-th window is denoted as $\mathbf{hFC}(k) = \left\{ hFC_{ij}(k) \right\}_{i \leqslant i,j \leqslant R}$, and the high-order dynamic FC network is represented as $\mathbf{hFC} = [hFC(1), hFC(2), \cdots, hFC(K)]$.

## Constructing dynamic weighted graph neural network

Since the whole brain FC network is a dense graph, it will bring difficulties to the training of GNN, and previous studies have shown that there are redundant FC features useless for disease classification in dynamic FC network. Therefore, we use $t$-test to extract features to remove redundant FC features and the FC matrix is a sparse matrix after $t$-test. Here, we use sparse FC matrix ($\mathbf{lFC}(k)$ or $\mathbf{hFC}(k)$) as the weighted adjacency matrix of the temporal $k$-th GNN and the weighted-graph local clustering coefficient [16] is used as the node feature of GNN. The weighted-graph local clustering coefficient for node (brain region) $i$ in $k$-th window is defined as:

$$
n_i(k) = \frac{2 \sum_{j:j \in \mathcal{N}_i} \left( FC_{ij}(k) \right)^{\frac{1}{3}}}{|\mathcal{N}_i| (|\mathcal{N}_i| - 1)}
\tag{3}
$$

where $\Delta_i$ denotes the set of nodes directly connected to node $i$, $|\Delta_i|$ is the number of elements in $\Delta_i$, and $FC_{ij}(k)$ is $lFC_{ij}(k)$ ($hFC_{ij}(k)$) when the brain network is low-order (high-order) dynamic FC network.

In this paper, we employ graph convolution network (GCN) [34] as the infrastructure of GNN. Each FC network is a weighted graph with NC or eMCI labels, so this study is based on the graph classification task. The propagation rule of a multi-layer GCN can be expressed as:

$$
H^{(l+1)}(k) = \sigma \left( \tilde{D}^{-\frac{1}{2}} \tilde{A} \tilde{D}^{-\frac{1}{2}} H^{(l)}(k) W^{(l)} \right)
\tag{4}
$$

where $l$ denotes the $l$-th layer, $\sigma(\cdot)$ is the activation function, $\mathbf{A} \in \mathbb{R}^{R \times R}$ is an adjacency matrix, $\tilde{\mathbf{A}} = \mathbf{A} + I_{116}$ (in our case, $\tilde{\mathbf{A}}$ is the temporal FC matrix), $\tilde{\mathbf{D}}$ is the degree matrix of $\tilde{A}$, and $\tilde{\mathbf{D}}_{ii} = \sum_j \tilde{\mathbf{A}}_{ij}$, $H$ is the node feature which is initialized as $H^0(k) = (n_1(k), \cdots, n_i(k), \cdots, n_R(k))$, $W$ is the learnable weight parameter. Fig 1(c) demonstrates the construction of temporal GNNs for temporal FC networks.

## Using LSTM to extract sequence features

After constructing the dynamic GNN network, we use the LSTM architectures to extract sequence information, and finally, use the fully connected layer for classification, as shown in Fig 1(d).

Specifically, the output of dynamic GCN is used as the input of LSTM. Let $H^{out}(k) = (n_1(k), \cdots, n_i(k), \cdots, n_R(k))$ represent the output of the $k$-th subnetwork of the dynamic GCN. Note that the input and output of GCN in this work are of the same dimension.

The detailed LSTM feature extraction diagram is shown in Fig 2. In the LSTM model, at each time step $k$, the forget gate $f_k$ determines the information to discard from the previous cell state, allowing the model to selectively forget irrelevant information. Simultaneously, the input gate $i_k$ and the candidate cell state $\hat{C}_k$ work together to update the cell state $C_k$ with new information relevant to the task at hand. The updated cell state and the output gate's activation then determine the final output, the hidden state $h_k$, which serves as input for the subsequent time step or the next layer in the network. Each LSTM cell operates as follows:

$$f_k = \sigma\left(W_f \cdot [h_{k-1}, h_k] + b_f\right) \tag{5}$$

$$i_k = \sigma\left(W_k \cdot [h_{k-1}, h_k] + b_i\right) \tag{6}$$

$$\tilde{C}_k = \tanh\left(W_C \cdot [h_{k-1}, h_k] + b_C\right) \tag{7}$$

$$C_k = f_k \cdot C_{k-1} + i_k \cdot \tilde{C}_k \tag{8}$$

$$o_k = \sigma\left(W_o [h_{k-1}, h_k] + b_o\right) \tag{9}$$

$$h_k = o_k \cdot \tanh\left(C_t\right) \tag{10}$$

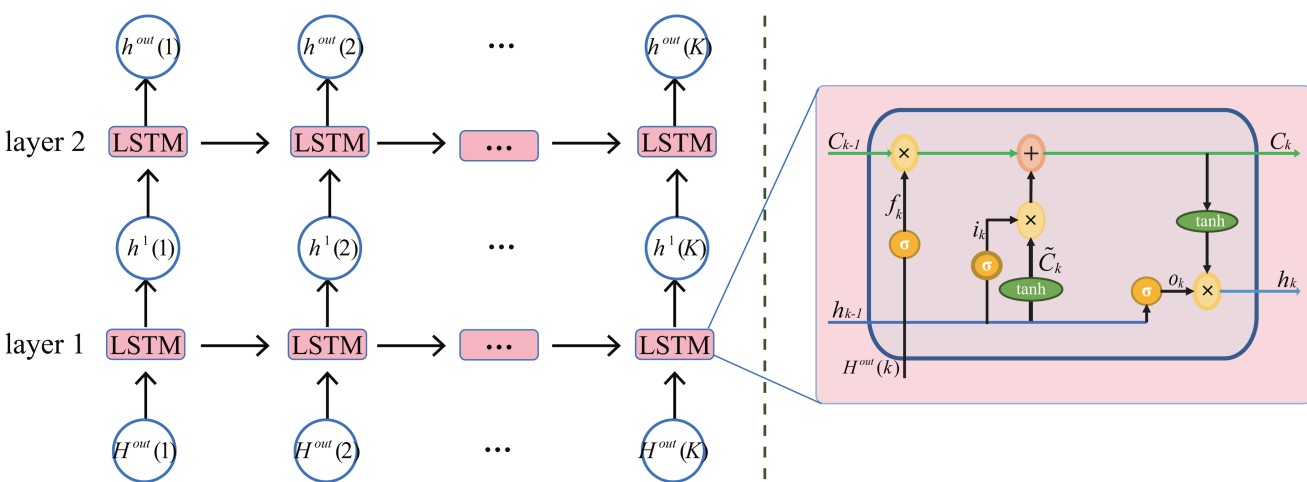

**Fig 2. The flowchart of LSTM for dynamic GNN.** On the left is the overall structure of LSTM, and on the right is the structure of LSTM cell.

where $b_f$ is the bias term, $W_f$ is a learnable parameter, and $h_k$ represents the $k$-th input of the $l$-th layer. For example, $h_k$ in the input layer is $H^{out}(k)$. Other symbols have similar meanings.

Finally, the features output by the LSTM network are input into the classifier (composed of fully connected layers) for classification.

## Results

### Dataset

In this paper, we used 319 unprocessed RS-fMRI data obtained from the ADNI database, including 154 normal controls (NCs) and 165 eMCI data. The demographic data of 319 subjects are summarized in Table 1, where the $p$-value was obtained by two-sample two-tailed $t$-test. Table 1 provides a comprehensive overview of the demographic and clinical characteristics of the participants, including age, gender distribution, total number of participants, MMSE scores, and APOE $\varepsilon$4 status. Each subject was scanned for 7 minutes. The in-plane imaging resolution of each scan was 2.29 to 3.31 mm, the slice thickness was 3.31 mm, the TE (echo time) was 30 ms, and the TR (repetition time) was 2.2 to 3.1 s (140 volumes generated). The RS-fMRI data were preprocessed using FSL FEAT software. Before preprocessing, the first 3 volumes were discarded for magnetization equilibrium. The RS-fMRI data were collected into 116 predefined brain regions using AAL atlas [35]. We refer to literatures [3,4] for data preprocessing, and see literatures [3,4] for more details.

### Experimental setup

The configuration of our experimental equipment is as follows: 11th Gen Intel(R) Core(TM) i9-11900H CPU, 32GB DDR4 RAM, NVIDIA RTX A2000 GPU, 1T SSD, Windows 11 operating system, NVIDIA CUDA Toolkit: 11.2, PyTorch 1.8.0 with Python 3.8.

To ensure the robustness and reproducibility of our findings, we meticulously optimized our model's hyperparameters, including setting an initial learning rate of 0.001 with adaptive adjustments, and selecting a batch size of 64 to balance computational efficiency with the stability of gradient updates. The Adam optimizer, recognized for its effectiveness in handling sparse gradients, was chosen to facilitate the training process. Additionally, to counteract overfitting, we incorporated L2 regularization and dropout techniques, with training monitored through a dedicated validation set to enable early stopping based on performance improvements.

In Eq 4, we use the *ReLU* activation function, i.e., $\sigma(\cdot) = ReLU(\cdot)$. The parameter settings of the sliding window follow the literature [2], i.e., we set $L = 30$, $s = 2$. Other settings include an epoch of 400, a learning rate of 0.001, two hidden layers for GCN, and three hidden layers for LSTM. We use the six evaluation indicators commonly used in the classification task as the evaluation indicators of this experiment, as follows.

$$TPR = \frac{TP}{TP + FN} \tag{11}$$

**Table 1. Demographic Information of the Subjects**

| Group | Total Number of Participants | Age (mean $\pm$ SD) | Gender (M/F) | MMSE (mean $\pm$ SD) | APOE $\varepsilon$4 Carriers (n/N) |
|-------|------------------------------|----------------------|--------------|------------------------|-------------------------------------|
| NC | 154 | 75.36 $\pm$ 6.16 | 67/87 | 29.1 $\pm$ 1.2 | 30/154 |
| eMCI | 165 | 72.03 $\pm$ 7.26 | 73/92 | 27.4 $\pm$ 2.1 | 45/165 |

$$TNR = \frac{TN}{TN + FP} \tag{12}$$

$$PPV = \frac{TP}{TP + FP} \tag{13}$$

$$NPV = \frac{TN}{TN + FN} \tag{14}$$

$$ACC = \frac{TP + TN}{TP + TN + FP + FN} \tag{15}$$

$$F1 - Score = \frac{2 \cdot TP}{2 \cdot TP + FN + FP} \tag{16}$$

where *TP*, *TN*, *FP*, and *FN* represent True Positive, True Negative, False Positive, and False Negative, respectively.

The proposed method is compared with the following methods:

- **Support Vector Machine (SVM).** SVM is a classical algorithm, which is widely used in FC network classification. In this paper, we feed FC features (i.e. edge features of brain graphs) into SVM as input features for classification.
- **temporal GNN (tGNN).** In [29], the authors proposed a LSTM-based temporal GNN for AD classification, which uses FC feature (edge feature of brain image) as the node feature of GNN.
- **Temporal dynamics learning (TDL).** In [36], Wang et al. proposed a method for simulating the dynamic properties of the brain's functional connectivity network based on RS-fMRI data and utilizing complex network analysis and temporal sequence analysis techniques.
- **Functional connectivity with deep learning (FCDL).** In [37], Zhang et al. proposed a method to detect autism using FC analysis combined with feature selection and deep learning. They used a feature selection algorithm to select the most relevant feature subset from the FC matrix, and then, a deep learning model is used to learn representations and automatic classification of FC features.

## Classification performance

In this section, we show the classification results when the *t*-test parameter is 0.05 (i.e., 95% confidence interval). Low-order FC networks refer to the direct temporal correlations between pairs of brain regions, where the functional connectivity is typically calculated based on simple pairwise correlations of resting-state fMRI signals. These networks represent the most immediate and direct interactions between regions of the brain.High-order FC networks go beyond pairwise correlations and involve higher-order interactions, such as the correlations between multiple brain regions simultaneously. These networks capture more complex patterns of interaction, reflecting the higher-level coordination between different brain areas. Tables 2 and 3 show the classification performance of low- and high-order FC network in three methods respectively. It can be seen from Tables 2 and 3 that our proposed method has achieved the best performance compared with the comparison methods. For low-order FC network, the accuracy of the proposed method reaches 91.67%, which is 8.34%, 6.67%, 3.34% and 5.00% higher than SVM, tGNN, TDL and FCDL, respectively. The F1-score of the proposed method is 10.57%, 8.08%, 4.81% and 6.66% higher than SVM, tGNN, TDL and FCDL, respectively, reaching 91.80%. Furthermore, with the exception of NPV, the remaining indicators exhibit higher values compared to the comparison method. This observation

**Table 2. Performance of features extracted from low-order FC network.**

|          | SVM    | tGNN   | TDL    | FCDL   | Ours      |
|----------|--------|--------|--------|--------|-----------|
| ACC      | 83.33% | 85.00% | 88.33% | 86.67% | **91.67%** |
| TPR      | 80.00% | 86.67% | 87.10% | 83.87% | **89.74%** |
| TNR      | 86.67% | 83.33% | 89.66% | 89.66% | **95.24%** |
| PPV      | 85.71% | 83.87% | 90.00% | 89.66% | **97.22%** |
| NPV      | 81.25% | 86.21% | 86.67% | 83.87% | 83.33%    |
| F1-Score | 82.76% | 85.25% | 88.52% | 86.67% | **93.33%** |

**Table 3. Performance of features extracted from high-order FC network.**

|          | SVM    | tGNN   | TDL    | FCDL   | Ours      |
|----------|--------|--------|--------|--------|-----------|
| ACC      | 71.67% | 76.66% | 73.33% | 75.00% | **78.33%** |
| TPR      | 70.00% | 70.00% | 73.33% | 74.19% | **76.67%** |
| TNR      | 73.33% | 83.33% | 68.75% | 70.97% | 80.00%    |
| PPV      | 72.41% | 80.77% | 68.75% | 71.88% | 79.31%    |
| NPV      | 70.97% | 73.53% | 73.33% | 73.33% | **77.42%** |
| F1-Score | 71.18% | 75.00% | 70.97% | 73.02% | **77.97%** |

clearly demonstrates the exceptional performance of our method in relation to low-order FC networks.

For high-order FC network, our method achieves 78.33% accuracy and 77.97% F1-Score. The accuracy is 6.66%, 1.67%, 5.00% and 3.33% higher than SVM, tGNN, TDL and FCDL, respectively. And F1-score is 6.79%, 2.97%, 7.00% and 4.95% higher than SVM, tGNN, TDL and FCDL, respectively. Moreover, for all other indicators, with the exception of PPV and TNR, our method outperforms the comparison method. These results clearly demonstrate the outstanding performance of our method in high-order FC networks.

## Most discriminative resting-state networks

The brain regions are grouped into six resting-state networks according to BrainNet Viewer [38], including default mode network (DMN), executive attention network (EAN), visual network (VN), sensorimotor network (SMN), subcortical regions (SR) and cerebellum (CER). In order to evaluate the impact of different resting-state networks on the classification results, the features from the specific resting-state network are removed, and then fed into the trained network for classification. The more accuracy decreases, the more important the resting state network is for the classification of eMCI, that is, it is more discriminative. Table 4 shows the classification accuracy after removing the specific features in the low- and the high-order FC network, respectively. In Table 4, the "R-DMN" indicates the classification result after removing the features in DMN, and other symbols have similar meanings.

It can be seen from Table 4 that the removal of a specific resting network has a certain degree of impact on the classification results, but the removal of different resting networks has

**Table 4. Accuracy of remove specific features from low- and high-order FC network.**

|            | R-DMN  | R-EAN  | R-VN   | R-SMN  | R-SR   | R-CER  |
|------------|--------|--------|--------|--------|--------|--------|
| low-order  | 81.67% | 86.67% | 83.33% | 86.67% | 88.33% | 85.00% |
| high-order | 65.00% | 71.67% | 76.67% | 70.00% | 66.67% | 73.33% |

different effects. This is because in the resting state, eMCI has different degrees of damage to the functions of different networks.

## Discussion

### Analysis of classification results

In section , the classification results are shown. It can be drawn that the proposed method is significantly better than the comparison methods for both low-order FC network and high-order FC network. It can be seen that the proposed method, tGNN and FCDL are based on neural network, which are superior to the classification performance of SVM and TDL. In addition, the classification performance of the proposed method is better than that of tGNN, which shows that it is advantageous to regard the weight of FC network as the weight of GNN edges.

An interesting phenomenon can be seen from Tables 2 and 3, that is, no matter which classifier, the classification results based on low-order FC network are significantly better than those based on high-order FC network. The possible reason is that the high-order FC network reflects the complex FC interaction pattern of brain regions, and blends the FC information of multiple brain regions together. This can produce better classification effect for some neurological diseases with obvious damage to brain function, such as autism spectrum disorder [39]. However, compared with NC, there is no obvious abnormality in brain function of eMCI patients, which is also the reason why the detection of eMCI is still an important challenge. Therefore, compared with the low-order FC network, the complex interaction pattern of the high-order FC network provides less discriminative features, resulting in poor classification effect.

### Analysis of discriminative resting-state networks

The brain regions in the six resting-state networks are responsible for different functions in the task-free state, such as DMN is known to activate at resting state and deactivate during task performance [40] and SMN is a large brain resting-state network, which is activated in sports tasks [41].

It can be seen from Table 4 that the removal of specific networks results in different degrees of degradation of classification performance. Especially when DMN is removed, the classification accuracy is significantly reduced, reducing the accuracy of 10.00% on the low-order FC network and 13.33% on the high-order FC network. This is because DMN is activated in the resting state, responsible for the main activities of the brain, and eMCI disrupts the function of DMN, that is, DMN has a functional lesion. This is because DMN is activated in the resting state, responsible for the main activities of the brain, and eMCI disrupts the function of DMN, that is, DMN has a functional lesion. Therefore, DMN is very important for the detection of eMCI and provides the discriminative features for the detection of eMCI. When VN is removed, the accuracy of low- and high-order FC network is reduced by 8.34% and 1.66%, respectively. This because VN is mainly responsible for the human visual system. eMCI causes the visual function of patients to be damaged [42], thus destroying the FC pattern in the brain regions of patients with VN. Therefore, VN is also important for the detection of eMCI. SMN is a large-scale brain resting-state network activated in motor tasks. When it is removed, the accuracy of low- and high-order FC networks is reduced by 5.00% and 8.33% respectively. For EAN, SR and CER, the accuracy is also reduced when they are removed. From Table 4, we can conclude that eMCI leads to different degrees of abnormality of FC mode in the whole brain.

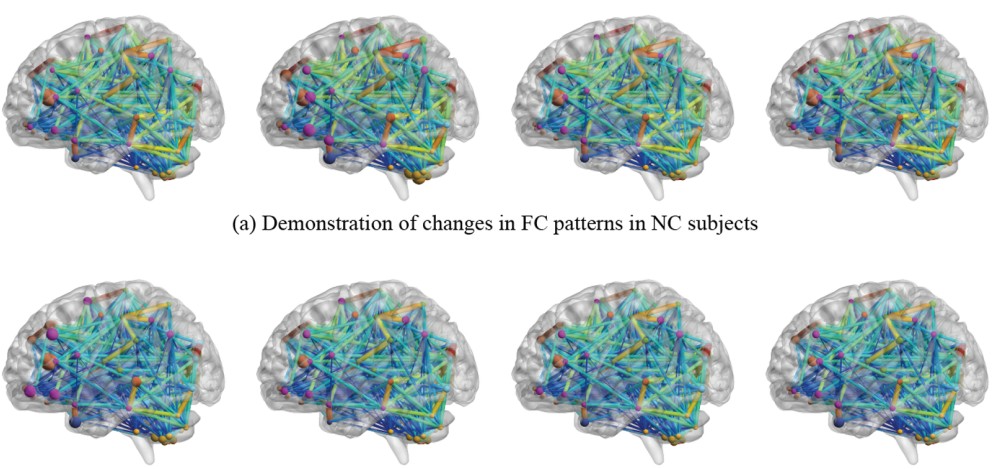

(a) Demonstration of changes in FC patterns in NC subjects

(a) Illustration of alterations in FC patterns in eMCI subjects

**Fig 3. The changing brain regions.** Note that we only display a partial representation of the FC subnetwork due to space constraints.

Fig 3 demonstrates the dynamic changes in brain regions of NC subjects and eMCI subjects. The thickness of the connecting lines depicts the strength of FC, while the solid circles represent node features. It should be noted that we obtained a total of 54 subgraphs (FC subnetworks), but due to space limitations we only show the 1st, 15th, 29th and 54th (from left to right) subgraphs. As can be seen from Fig 3, the brain regions of NC subjects did not change much. This is because the neural activity of normal people is not active in the resting state. However, patients with eMCI have abnormal brain activity in the resting state, resulting in abnormal FC networks. Specifically, regions such as the dorsolateral prefrontal cortex and posterior cingulate cortex showed significantly altered connectivity patterns in eMCI patients compared to NC subjects [43,44]. These aberrant connectivity patterns indicate disturbances in DMN, which plays a crucial role in self-referential thinking and memory consolidation [45]. Moreover, the disrupted FC in eMCI patients can further lead to cognitive impairment and decline in various domains such as attention, memory, and executive functions. Understanding these aberrant brain activity patterns and their impact on cognitive function in eMCI patients is crucial for early detection and intervention strategies. Therefore, considering the dynamic changes in brain FC patterns can play an important role in detecting eMCI, which is also an important reason for the success of our model.

## Conclusion

In this paper, we propose a new perspective that the weight of FC network is used as the weight of the adjacency matrix of GNN, and the local clustering coefficient of the weighted graph of nodes is used as the input feature of GNN. Our approach stands apart from previous methods, which typically treat FC features as GNN node attributes while disregarding the topological characteristics specific to FC networks. In contrast, our method successfully combines the FC topological characteristics with GNN graph properties, achieving a unified and comprehensive framework. We use ADNI dataset to carry out experiments on two types of FC networks (i.e., low-order FC network and high-order FC network), and the results demonstrate that the proposed method is effective and superior to the comparison methods.

## Future directions

While the current study focused on the detection of eMCI as a precursor to Alzheimer's Disease, we acknowledge the complexity and heterogeneity of AD manifestations across different subtypes. Future work will aim to extend our methodology to include a wider range of AD subtypes, such as late mild cognitive impairment (lMCI), Alzheimer's Disease Dementia (ADD), and other forms of neurodegenerative conditions. This expansion will allow us to explore the unique topological and dynamic properties of functional connectivity networks associated with each subtype. By doing so, we hope to develop more nuanced models that can accurately identify and differentiate between various stages and forms of Alzheimer's Disease, ultimately contributing to personalized diagnostic tools and therapeutic strategies.

## Author contributions

**Conceptualization:** Li Liu, Kai Yang.

**Funding acquisition:** Li Liu.

**Methodology:** Yifei Li.

**Software:** Yifei Li.

**Validation:** Kai Yang.

**Writing – original draft:** Li Liu, Kai Yang.

**Writing – review & editing:** Kai Yang.

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
