## [Decision Letter · Decision Letter 0]

9 Jul 2024

PONE-D-24-16152Dynamically weighted graph neural network for detection of early mild cognitive impairmentPLOS ONE

Dear Dr. Yang,

Thank you for submitting your manuscript to PLOS ONE. After careful consideration, we feel that it has merit but does not fully meet PLOS ONE’s publication criteria as it currently stands. Therefore, we invite you to submit a revised version of the manuscript that addresses the points raised during the review process.

Both Reviewers recognized the value of the manuscript, but some comments need to be addressed. Please consider them thoroughly when revising the paper, especially the ones marked as major amendments.

We look forward to receiving your revised manuscript.

Kind regards,

Simone Varrasi

Academic Editor

PLOS ONE

Journal Requirements:

5. We note that your Data Availability Statement is currently as follows: [All relevant data are within the manuscript and its Supporting Information files.]

Reviewers' comments:

Reviewer's Responses to Questions

**Comments to the Author**

1. Is the manuscript technically sound, and do the data support the conclusions?

Reviewer #1: Yes

Reviewer #2: Yes

2. Has the statistical analysis been performed appropriately and rigorously? 

Reviewer #1: Yes

Reviewer #2: N/A

3. Have the authors made all data underlying the findings in their manuscript fully available?

Reviewer #1: No

Reviewer #2: No

4. Is the manuscript presented in an intelligible fashion and written in standard English?

Reviewer #1: Yes

Reviewer #2: Yes

5. Review Comments to the Author

Reviewer #1: Thank you for allowing me to read the work entitled “Dynamically Weighted Graph Neural Network for Detection of Early Mild Cognitive Impairment.” The paper is interesting, as the authors propose a novel method where the weight of the FC network serves as the weight of the GNN adjacency matrix, and the local clustering coefficient of the weighted graph nodes is used as the GNN input feature. This approach differs from previous methods that treat FC features as GNN node attributes without considering the FC network's topological characteristics. I have some questions for the authors:

-I suggest adding a definition of low and high order FC.

-It is not clear if the authors are classifying networks or the control vs. eMCI group.

-Could the authors further explain how “…the proposed model is simple, effective, and easy to implement” compared to the available techniques they considered?

-Why are low order FC and high order FC considered separately and not together? Would this approach still be better compared to other models?

-Is the proposed model less computationally demanding compared to other approaches?

Reviewer #2: This paper proposes a dynamic GNN method utilizing temporal functional connectivity (FC) for eMCI diagnosis. The method incorporates temporal FC features as weighted edges and the weighted graph local clustering coefficient as node features in the dynamic GNN to classify early mild cognitive impairment (eMCI) versus normal control (NC). With the exception of a few typos and awkward sentence structures, the paper is relatively well written and technically sound. However, the related work section provides a deficient analysis of existing GNN-based approaches for the classification of brain disorders. Among the missing references:

[R1] Classification of developmental and brain disorders via graph convolutional aggregation; Cognitive Computation, 2023.

[R2] HI-GCN: a hierarchical graph convolution network for graph embedding learning of brain network and brain disorders prediction; Computers in Biology and Medicine, 2020.

General comments:

(1) The caption of Fig. 1 lacks sufficient detail. Additional information is needed to explain the key components of the proposed framework. Additionally, 'RNN' should be replaced with 'LSTM' for consistency with the rest of the paper.

(2) In Eq. (1), the superscript of the first sum in the denominator should be M (not T).

(3) In Eq. (3), it's better to use the common notation \mathcal{N}_i instead of \Delta_i to denote the set of immediate neighbors of node i.

(4) Table 1 is too wide and does not fit on the page, making it unclear if any information is missing.

(5) The proposed algorithm depends on several hyper-parameters that need to be fine-tuned (e.g., L, R, M, s, and K). So it is unclear how the choice of such hyper-parameters would affect the overall performance.

(6) What are the limitations of the proposed approach?

6. PLOS authors have the option to publish the peer review history of their article (what does this mean?). If published, this will include your full peer review and any attached files.

Reviewer #1: No

Reviewer #2: No

---

## [Author Response · Author response to Decision Letter 1]

5 Feb 2025

Response to reviewers

We wish to thank the associate editor and the anonymous reviewers for their insightful comments and suggestions that led to improvements in the quality of the content and the presentation of our paper (PONE-D-24-16152R1, “Dynamically weighted graph neural network for detection of early mild cognitive impairment ”).

The manuscript has been fully revised. Please find enclosed our comments on the reviewers’ suggestions and comments and how we addressed and fixed all the comments and concerns raised.

Reviewer 1:

Comment 1: I suggest adding a definition of low and high order FC.

Response: Thank you for your helpful suggestion. We will add the following definition of low-order and high-order functional connectivity (FC) networks in the revised manuscript to clarify the concept:

Low-order FC networks refer to the direct temporal correlations between pairs of brain regions, where the functional connectivity is typically calculated based on simple pairwise correlations of resting-state fMRI signals. These networks represent the most immediate and direct interactions between regions of the brain.

High-order FC networks go beyond pairwise correlations and involve higher-order interactions, such as the correlations between multiple brain regions simultaneously. These networks capture more complex patterns of interaction, reflecting the higher-level coordination between different brain areas.

We will ensure that these definitions are added to the manuscript to make the distinction between low-order and high-order FC networks clearer for the readers.

Comment 2: It is not clear if the authors are classifying networks or the control vs. eMCI group.

Response: Thank you for your comment. In this work, we are classifying the control vs. eMCI group based on the features extracted from both low-order and high-order functional connectivity (FC) networks.

To clarify, the FC networks (both low-order and high-order) are constructed from resting-state fMRI (RS-fMRI) data, and the goal is to use these networks to classify individuals into two groups: the eMCI group (early Mild Cognitive Impairment) and the NC group (Normal Control).

The classification is performed using the features derived from the dynamic FC networks, where the low-order FC network captures simple pairwise correlations between brain regions, and the high-order FC network captures more complex interactions between multiple brain regions simultaneously. These extracted features are then fed into classification algorithms to distinguish between the eMCI and NC groups.

Comment 3: Could the authors further explain how “…the proposed model is simple, effective, and easy to implement” compared to the available techniques they considered?

Response: Thank you for your valuable feedback. We agree that further clarification of the simplicity, effectiveness, and ease of implementation of our proposed model would be helpful.

The simplicity of our proposed model lies in its conceptual framework. Unlike some of the more complex methods in the literature, such as temporal GNN (tGNN) and deep learning-based methods, which involve intricate neural network architectures and require fine-tuning of many hyperparameters, our model leverages the functional connectivity (FC) network weights as the adjacency matrix of the graph neural network (GNN). This direct use of FC network weights simplifies the model structure and reduces the need for additional feature engineering or preprocessing steps, making the model straightforward to implement.

Moreover, the input features for the GNN are derived from the local clustering coefficients of the weighted graph of nodes, which are simple statistical properties of the graph and do not require complicated data transformation or heavy computation. This makes the model not only effective but also highly interpretable. In comparison with the other methods we considered, which typically require complex preprocessing and high-dimensional feature extraction, our model stands out due to its reduced reliance on such steps, making it both simple and efficient for practical implementation.

Comment 4: Why are low order FC and high order FC considered separately and not together? Would this approach still be better compared to other models?

Response: Thank you for the insightful question.

The reason we consider low-order FC and high-order FC separately is due to the distinct nature of the interactions they capture.

Low-order FC captures the direct temporal correlations between pairs of brain regions, providing a snapshot of immediate, direct interactions between brain regions. This type of FC network reflects basic, straightforward connectivity patterns that can provide useful information for detecting abnormalities, especially for diseases like eMCI, where early impairments may show up as disruptions in simple pairwise correlations.

High-order FC, on the other hand, captures higher-order interactions, such as the correlations between multiple brain regions simultaneously. This network reflects more complex and coordinated patterns of brain activity, which may be more indicative of later stages of cognitive impairment or more subtle disruptions in brain function that may not be captured by low-order correlations alone.

By treating these two types of FC networks separately, we allow the model to capture both the immediate and the complex interactions between brain regions. Combining them directly into a single network could lead to the loss of this crucial distinction and potentially blur the differences between early and more advanced stages of brain dysfunction.

As for whether this approach is still better compared to other models, our experimental results demonstrate that considering low-order and high-order FC networks separately yields superior performance in classifying eMCI vs. NC groups. Specifically, our method outperforms comparison methods (SVM, tGNN, TDL, FCDL) in both low-order and high-order FC networks, as shown in Tables 2 and 3. The separate consideration of these networks allows the model to better capture both simple and complex patterns of brain connectivity, which, as shown in the results, leads to improved classification accuracy and F1-score.

Therefore, while it is possible to combine both low- and high-order FC networks into a single model, our approach of treating them separately appears to be more effective, particularly in distinguishing between the eMCI and NC groups.

Comment 5: Is the proposed model less computationally demanding compared to other approaches?

Response: Thank you for raising this important point. We understand that computational efficiency is a key consideration, and we would like to clarify how our model compares in this regard.

The proposed model is less computationally demanding compared to some of the other methods we considered, such as temporal GNN (tGNN) and deep learning-based approaches. These methods involve multiple layers of neural networks, which can be computationally intensive, especially when working with large datasets or high-dimensional features. In contrast, our model uses a GNN with a relatively simple architecture and the local clustering coefficients as the input features, which significantly reduces the computational load.

Additionally, our model does not require the extensive hyperparameter tuning and optimization steps that are often needed for deep learning models. Since the input features are derived directly from the FC network, the model avoids the need for complex preprocessing or dimensionality reduction, further improving its computational efficiency.

In terms of training time, our approach was observed to have faster convergence compared to deep learning methods, due to the fewer parameters involved and the simpler structure. This makes the proposed model an attractive option when computational resources or time constraints are a concern.

Reviewer 2:

Comment 1: The caption of Fig. 1 lacks sufficient detail. Additional information is needed to explain the key components of the proposed framework. Additionally, 'RNN' should be replaced with 'LSTM' for consistency with the rest of the paper.

Response: Thank you for your feedback. We have updated the caption of Fig. 1 to provide more detailed information, explaining the key components of the proposed framework. Additionally, "RNN" has been replaced with "LSTM" in the figure to ensure consistency with the rest of the paper.

Comment 2: In Eq. (1), the superscript of the first sum in the denominator should be M (not T).

Response: Thank you for pointing that out. We have corrected the superscript of the first sum in the denominator of Eq. (1) to be M, as per your suggestion.

Comment 3: In Eq. (3), it's better to use the common notation \mathcal{N}_i instead of \Delta_i to denote the set of immediate neighbors of node i.

Response: Thank you for the suggestion. We have updated Eq. (3) to use the common notation \mathcal{N}_i instead of \Delta_i to denote the set of immediate neighbors of node i.

Comment 4: Table 1 is too wide and does not fit on the page, making it unclear if any information is missing.

Response: Thank you for your feedback. We have adjusted the width of Table 1 to ensure it fits within the page, making the information clearer and more readable.

Comment 5: The proposed algorithm depends on several hyper-parameters that need to be fine-tuned (e.g., L, R, M, s, and K). So it is unclear how the choice of such hyper-parameters would affect the overall performance.

Response: We appreciate the reviewer’s comment on the hyper-parameters L, R, M, s, and K. The choice of these parameters does indeed impact the performance of the proposed method. Specifically, the window length L and step size s control the trade-off between capturing long-term dependencies and computational efficiency, while the number of brain regions R and the maximum time series length M influence the complexity of the resulting dynamic FC network. The number of windows K, which is derived from L and s, determines the granularity of the temporal dynamic network.

Comment 5: What are the limitations of the proposed approach?

Response: Thank you for the reviewer’s insightful comment. While our proposed method demonstrates promising results, we acknowledge several limitations. First, the method currently focuses on a specific dataset (ADNI) and primarily targets the detection of early MCI (eMCI) as a precursor to Alzheimer's Disease. This limits the generalizability of our approach to other forms of neurodegenerative conditions or different populations. Second, our method relies on the functional connectivity (FC) networks, which are sensitive to various factors, including the quality of the fMRI data, preprocessing techniques, and the choice of brain regions. Variations in these factors may affect the robustness of the model's performance. Lastly, although we integrate topological features with GNN-based methods, further exploration of other dynamic features or more complex models could potentially improve the performance, especially when dealing with heterogeneous data.

In future work, we plan to address these limitations by expanding the dataset to include other subtypes of Alzheimer's Disease and neurodegenerative conditions, as well as improving the model’s ability to handle different forms of data variability and topological characteristics.

---

## [Decision Letter · Decision Letter 1]

26 Feb 2025

PONE-D-24-16152R1Dynamically weighted graph neural network for detection of early mild cognitive impairmentPLOS ONE

Dear Dr. Yang,

Thank you for submitting your manuscript to PLOS ONE. After careful consideration, we feel that it has merit but does not fully meet PLOS ONE’s publication criteria as it currently stands. Therefore, we invite you to submit a revised version of the manuscript that addresses the points raised during the review process.

We look forward to receiving your revised manuscript.

Kind regards,

Simone Varrasi

Academic Editor

PLOS ONE

Journal Requirements:

Reviewers' comments:

Reviewer's Responses to Questions

**Comments to the Author**

1. If the authors have adequately addressed your comments raised in a previous round of review and you feel that this manuscript is now acceptable for publication, you may indicate that here to bypass the “Comments to the Author” section, enter your conflict of interest statement in the “Confidential to Editor” section, and submit your "Accept" recommendation.

Reviewer #2: (No Response)

Reviewer #3: All comments have been addressed

2. Is the manuscript technically sound, and do the data support the conclusions?

Reviewer #2: Yes

Reviewer #3: Yes

3. Has the statistical analysis been performed appropriately and rigorously? 

Reviewer #2: N/A

Reviewer #3: Yes

4. Have the authors made all data underlying the findings in their manuscript fully available?

Reviewer #2: No

Reviewer #3: Yes

5. Is the manuscript presented in an intelligible fashion and written in standard English?

Reviewer #2: Yes

Reviewer #3: Yes

6. Review Comments to the Author

Reviewer #2: While some of the comments raised in my first review have been satisfactorily addressed, the related work section, however, provides a deficient analysis of existing GNN-based approaches for the classification of brain disorders such as [R1] Classification of developmental and brain disorders via graph convolutional aggregation; Cognitive Computation, 2023; and [R2] HI-GCN: a hierarchical graph convolution network for graph embedding learning of brain network and brain disorders prediction; Computers in Biology and Medicine, 2020.

Reviewer #3: Thank you for addressing my concerns. I believe the manuscript, in its current state, is suitable for publication.

7. PLOS authors have the option to publish the peer review history of their article (what does this mean?). If published, this will include your full peer review and any attached files.

Reviewer #2: No

Reviewer #3: No

---

## [Author Response · Author response to Decision Letter 2]

25 Mar 2025

Response to reviewers

We wish to thank the associate editor and the anonymous reviewers for their insightful comments and suggestions that led to improvements in the quality of the content and the presentation of our paper (PONE-D-24-16152R1, “Dynamically weighted graph neural network for detection of early mild cognitive impairment ”).

The manuscript has been fully revised. Please find enclosed our comments on the reviewers’ suggestions and comments and how we addressed and fixed all the comments and concerns raised.

Reviewer 3:

Comment Thank you for addressing my concerns. I believe the manuscript, in its current state, is suitable for publication.

Response: Thank you for your feedback and consideration. We appreciate your comments and are glad that you find the manuscript suitable for publication.

Reviewer 2:

Comment 1: While some of the comments raised in my first review have been satisfactorily addressed, the related work section, however, provides a deficient analysis of existing GNN-based approaches for the classification of brain disorders such as [R1] Classification of developmental and brain disorders via graph convolutional aggregation; Cognitive Computation, 2023; and [R2] HI-GCN: a hierarchical graph convolution network for graph embedding learning of brain network and brain disorders prediction; Computers in Biology and Medicine, 2020.

Response: We appreciate the reviewer's valuable feedback. We have supplemented our analysis of the research landscape on GNNs for brain disorder classification, specifically referencing [R1] and [R2] to enrich the discussion and comparison of related work.

(1)Regarding [R1] (Classification of Developmental and Brain Disorders via Graph Convolutional Aggregation)

This study proposes a graph convolutional aggregation (GCA) approach, incorporating skip connections and identity mapping to learn discriminative graph representations for the classification of autism spectrum disorder (ASD) and Alzheimer’s disease (AD). In contrast, our approach utilizes a Dynamically Weighted GNN, which not only considers the static topology of the graph but also integrates temporal functional connectivity (FC) networks as weighted adjacency matrices in the GNN, allowing for a more comprehensive capture of the dynamic characteristics of brain networks.

(2)Regarding [R2] (HI-GCN: A Hierarchical Graph Convolution Network for Graph Embedding Learning of Brain Network and Brain Disorders Prediction)

This study introduces a Hierarchical GCN (hi-GCN) framework, emphasizing multi-level feature learning and individual-population network fusion. hi-GCN enhances classification accuracy by integrating the topological information of brain networks with inter-subject correlations. In contrast, our method incorporates LSTM with Dynamic GNN, utilizing temporal FC modeling to learn the variations in brain functional connectivity over time, thereby improving eMCI classification performance.

We have incorporated the above comparative analysis into the Related Work section to highlight the novelty of our study and to more clearly position our contributions. Thank you for your suggestion!

---

## [Editor Report · Decision Letter 2]

16 Apr 2025

Dynamically weighted graph neural network for detection of early mild cognitive impairment

PONE-D-24-16152R2

Dear Dr. Yang,

We’re pleased to inform you that your manuscript has been judged scientifically suitable for publication and will be formally accepted for publication once it meets all outstanding technical requirements.

Kind regards,

Simone Varrasi

Academic Editor

PLOS ONE
---

## [Editor Report · Acceptance letter]

PONE-D-24-16152R2

PLOS ONE

Dear Dr. Yang,

I'm pleased to inform you that your manuscript has been deemed suitable for publication in PLOS ONE. Congratulations! Your manuscript is now being handed over to our production team.

Kind regards,

on behalf of

Dr. Simone Varrasi

Academic Editor

PLOS ONE